# Temperature Evolution inside Hollow Core Wood Elements and Fire Resistance

**Domingos Pereira, Elza M. M. Fonseca *** and **Miguel Osório**

ISEP, Instituto Politécnico do Porto, R. Dr. António Bernardino de Almeida, 4249-015 Porto, Portugal;
1180819@isep.ipp.pt (D.P.); 1180724@isep.ipp.pt (M.O.)
***** Correspondence: elz@isep.ipp.pt

**Abstract:** The present study is focused on wall panels exposed to fire, with the construction building elements we used being made of wood and gypsum board materials. This type of configuration forms hollow core wood due to the constructive process. The aim is to present a numerical study to approach the calculation of the temperature inside hollow core wood elements and measure their fire resistance. The temperature evolution inside the cavities will be obtained with recourses to obtain the heat effect by convection and radiation through the wall elements. A numerical model, previously validated by the authors, will be used to carry out this process. The methodology includes the use of the finite element method in thermal and transient analysis with nonlinear materials to calculate temperature. To measure the fire resistance of the constructive model, the thermal insulation criterion, defined by the EN 1363-1:2020 standard, will be applied. Different results will be presented to discuss and ensure the verification of these fire-resistant elements.

**Keywords:** hollow core; wood; fire; fire resistance; thermal insulation criterion

## 1. Introduction

Buildings constructed with wood are becoming more prevalent worldwide due to the sustainability opportunities for wood construction and the general aesthetics of a wood building [1]. Wall panels with hollow wood core as lightweight elements are available for many applications [1,2]. Due to hollow core's high thermal mass, storing, and releasing heat, it maintains the ability to efficiently transfer large amounts of heat. The hollow wood core is a cavity within the component in use that offers some design flexibility. Another important factor is that hollow core wall panels have some advantages in relation to environmental impact in order to maintain sustainability and environmental friendliness, such as a reduced amount of cement and water required for production; no need for on-site storage space; minimal debris and site disturbance; and reduced labor, material, and waste.

Fire-resistance rating is the period, in minutes, for which the constructive element maintains the ability to confine a fire, to continue to perform a given structural function, or both, as determined by tests, or methods based on them, or more generally the time at which the failure of the element occurs [3]. It is defined by measuring the ability of a passive fire protection material to resist a standard fire resistance test. It comprises the time during which three criteria are satisfied: structural adequacy (ability to maintain stability and adequate load-bearing capacity), integrity (ability to resist the passage of flames and hot gases), and insulation (ability to maintain a temperature over the whole of the unexposed surface below that specified) [3–6]. With the increasing use of wood structures in construction engineering, more designers and engineers require information on detailed specifications and compliance with codes and regulations. There is still no complete certainty about the requirements or necessary details, namely focusing on the design of the structure in accidental situations and the calculation of fire resistance.

In this work, a typical model used in construction will be studied, such as walls, partitions, barriers, and corridors, to guarantee safety in their application in fire situations.

For example, according to the international building code [3], fire partitions must have a fire-resistance rating of no less than 1 h. There are exceptions, for example, corridor walls may have a half-hour fire-resistance rating. In the present work, fire resistance will be measured using the constructive model by applying the thermal insulation criterion, defined by the EN 1363-1:2020 standard [5]. A numerical methodology using ANSYS® [7] will be used to present a typical finite element mesh that can be used to calculate the effect on the internal elements of the hollow core by calculating the evolution of the internal temperature. This numerical methodology was verified by the authors in previous research [8–10] when comparing the developed numerical model with the experimental studies presented by H. Takeda and J. R. Mehaffey [11], which present different results in small- and full-scale fire resistance tests. In this work, different numerical models will be presented to control some geometric parameters that can affect the temperature evolution in these types of constructive models. Discussion of the results will include issues of allowable fire resistance for use in typical hollow core wood elements.

## 2. Model Description

Figure 1 represents the typical constructive element of the study, applicable as a wall partition. This element is formed by columns in wood GL32H, protected by gypsum board type F on each side. One side will be exposed to fire and the other is unexposed. The steel connectors considered in the construction are not included in the numerical model.

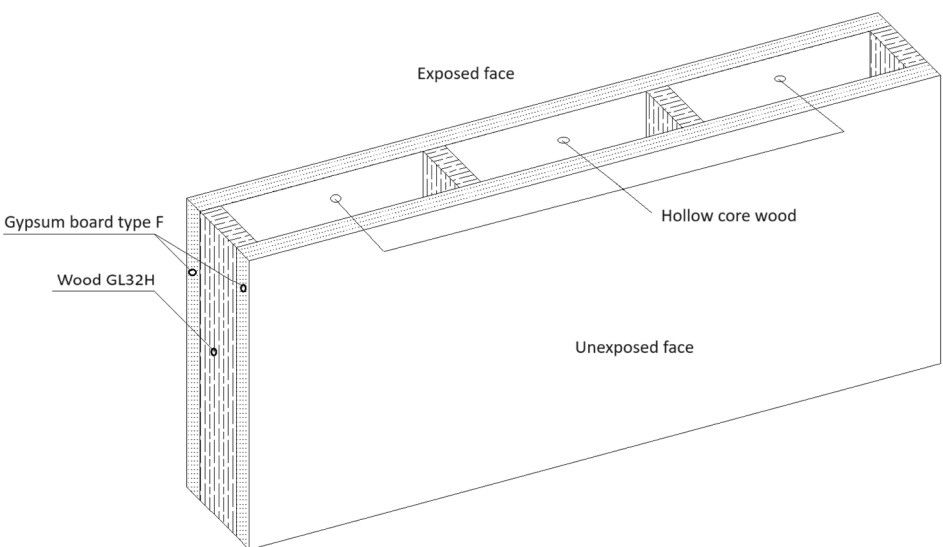

**Figure 1.** Wall partition in wood and gypsum board materials, with one side exposed to fire.

Due to the behavior of the constructive element, a cross-section in the two-dimensional (2D) plane will be considered in the analysis.

## 3. Thermal and Numerical Model

The ANSYS® program [7] was used as a finite element method for transient and nonlinear material thermal analysis [8,9]. A 2D mesh was used, representing a typical plane of the studied model. According to the used thermal materials, wood GL32H and gypsum board type F were obtained from the following references [12–14], as shown in Figure 2, in a particular hollow core of the studied models. The mesh size was determined considering the thin side, that is, the gypsum board with a thickness equal to 12.5 mm or 15 mm. In these zones, the number of finite elements was adjusted to two (less than 10 mm), always considering a mesh generated by ANSYS® [7] with a size of 10 mm. For the gypsum board, with a thickness of 25 mm, three finite elements were generated.

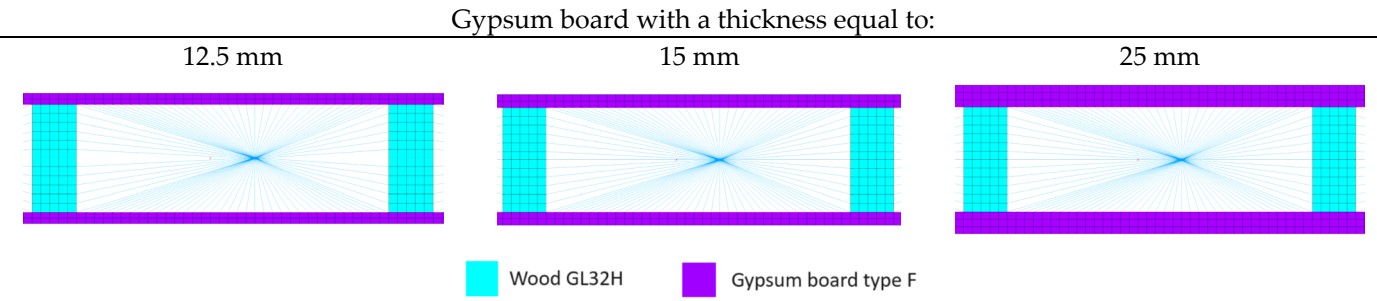

**Figure 2.** Hollow core element, mesh, and materials. The light blue color represents the wood, the violet color represents the gypsum board, and the shaded blue outlines the cavity.

The developed thermal model considers the variation in the material properties in terms of temperature. In summary, the thermal material properties are presented in Tables 1 and 2. The wood material was considered in glulam GL32H and gypsum board type F, both with a density equal to 480 kg/m$^3$ and 889 kg/m$^3$ at ambient temperature taken as 20 °C, respectively, as considered in a previous publication [12–14].

**Table 1.** Thermal properties of gypsum type F.

| Temperature, °C | Density, kg/m$^3$ | Temperature, °C | Specific Heat, kJ/KgK | Temperature, °C | Thermal Conductivity, W/mK |
|---|---|---|---|---|---|
| 20 | 889.00 | 20 | 0.95 | 20 | 0.190 |
| 100 | 889.00 | 100 | 0.95 | 195 | 0.190 |
| 170 | 737.87 | 135 | 25.00 | 155 | 0.100 |
| 600 | 737.87 | 170 | 0.95 | 200 | 0.103 |
| 750 | 700.98 | 650 | 0.95 | 400 | 0.113 |
| 1200 | 700.98 | 675 | 10.00 | 600 | 0.127 |
| | | 700 | 0.95 | 800 | 0.145 |
| | | 1200 | 0.95 | 1200 | 0.165 |

**Table 2.** Thermal properties of wood.

| Temperature, °C | Density, kg/m$^3$ | Temperature, °C | Specific Heat, kJ/KgK | Temperature, °C | Thermal Conductivity, W/mK |
|---|---|---|---|---|---|
| 20 | (1 + $w$) * 537.6 | 20 | 1.53 | 20 | 0.12 |
| 99 | (1 + $w$) * 537.6 | 99 | 1.77 | 200 | 0.15 |
| 120 | (1.0) * 480.0 | 100 | 13.60 | 350 | 0.07 |
| 200 | (1.0) * 480.0 | 120 | 13.50 | 500 | 0.09 |
| 250 | (0.93) * 446.4 | 121 | 2.12 | 800 | 0.35 |
| 300 | (0.76) * 364.8 | 200 | 2.00 | 1200 | 1.50 |
| 350 | (0.52) * 249.6 | 250 | 1.62 | | |
| 400 | (0.38) * 182.4 | 300 | 0.71 | | |
| 600 | (0.28) * 134.4 | 350 | 0.85 | | |
| 800 | (0.26) * 124.8 | 400 | 1.00 | | |
| 1200 | (0) * 0 | 600 | 1.40 | | |
| | * Density ratio | 800 | 1.65 | | |
| | $w$ moisture content | 1200 | 1.65 | | |

For standard fire exposure, the values of the thermal properties can be taken as proposed by EC5-1-2, EN 1995-1-2 (2003) [13]. This standard presents the variation in the wood thermal material depending on temperature, where density is a function of the moisture content, as a ratio presented in Table 2. The initial moisture content was equal to 12%, as proposed in Table 2, referring to [13].

In all numerical models, perfect contact between all the materials was assumed to allow for thermal energy conduction between them. The wood columns are represented in blue, and the gypsum board is in purple, as in Figures 2 and 3. The 2D numerical model has already been validated [8] considering the following elements: PLANE55, SURF151, and LINK34, as shown in Figure 3.

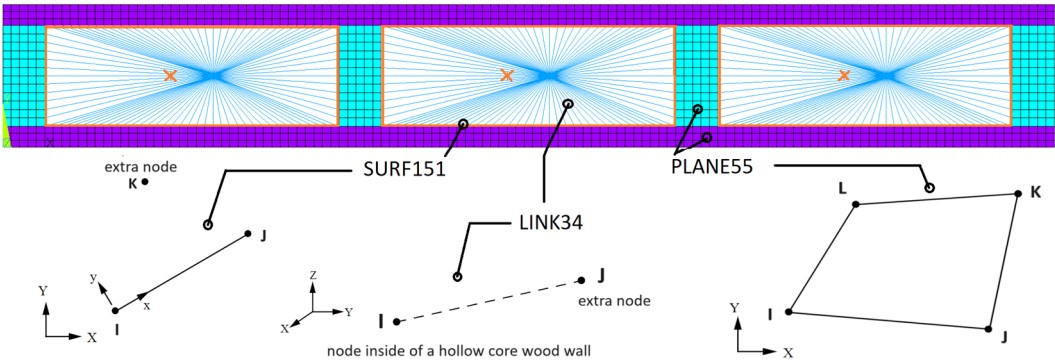

**Figure 3.** Two-dimensional mesh and finite elements. The light blue color represents the wood, the violet color represents the gypsum board, the orange color is the edge of the surface, and the shaded blue outlines the cavity.

PLANE55 is a 2D thermal solid element and can be used as a plane element for thermal conduction capacity. The element has four nodes with a single degree of freedom, temperature, at each node. The element is appropriate for 2D thermal analysis in steady-state or transient conditions [7,8].

SURF151 is a surface thermal element used for various load and surface effect applications. It may be overlaid on the face of any 2D thermal solid element. It allows for radiation between the internal surfaces of the cavity and any point inside it. The element consists of two nodes, associated with an extra node, located inside the cavity [7,8].

LINK34 is a uniaxial convection link element with the capacity to conduct heat between its nodes applicable to 2D or 3D thermal analysis in steady-state or transient conditions. The element has a single degree of freedom, temperature, at each node. The element is defined by two nodes, one node on the surface and another inside the hollow core, to represent a convection surface area [7,8].

The number of elements and nodes can lead to a more accurate solution. The decision on the number of finite elements, through a convergence process, and even through mesh control, is a function of each model. In this work, regarding mesh discretization, a finite element size of 10 mm was considered, where the mesh control is a function of the geometry to obtain a homogeneous and reasonable number of elements, as explained previously and shown in Figure 2. The boundary conditions included in the numerical model comply with Eurocode 1 part 1–2 [15], which allows for a fire effect to be induced in the constructive model. The boundary conditions are as follows:

-   Radiation and convection on the surface exposed to fire, with the effect of increasing temperature through the standard fire curve ISO 834 [16];
-   Only convection on the unexposed face;
-   Adiabatic conditions at the lateral edges;
-   The initial condition of the model corresponds to an ambient temperature of 20 °C.

Under these conditions, the relative emissivity of the wood material was 0.8 and 0.85 for gypsum board. The emissivity of the environment is a constant value and equal to 1,

with the convection coefficient equal to 25 W/m²K on the exposed face and 9 W/m²K on the unexposed face, according to Eurocode 1 part 1–2 [8,15]. Inside wood with a hollow core, an emissivity equal to 0.8 and a convection coefficient of 15.5 W/m²K were considered [8]. To satisfy the nonlinear conditions of the problem, the ANSYS® program [7] uses the Newton–Raphson iterative method, with a convergence criterion based on the heat flow with a tolerance equal to 0.9, and other variables being considered as standard parameters. At the end of the thermal analysis, the temperature evolution is obtained [8,9].

## 4. Results and Discussion of the Studied Hollow Core Wood Elements

### 4.1. Fire Resistance: Thermal Insulation Criterion

To obtain fire resistance, it is necessary to apply the thermal insulation criterion, defined by the EN 1363-1:2020 standard [5]. According to this standard, the elements fail when the insulation criterion is reached, regarding the average ($T_{ave}$) or maximum ($T_{max}$) temperature requirements on the unexposed face, above an initial temperature of $T_0 = 20\ °C$, displayed in Equations (1) and (2), respectively,

$$T_{ave} = T_0 + 140 \tag{1}$$

$$T_{max} = T_0 + 180 \tag{2}$$

To calculate fire resistance, it is necessary to obtain the temperature history at different points on the unexposed side. Thirteen nodal points were considered, as represented in Figure 4, to obtain the average and maximum temperature, which complies with the thermal insulation criterion, according to EN1363-1:2020 [5]. Furthermore, Figure 4 represents a central nodal point (K) in the hollow core element where the temperature inside the cavity will be determined. To verify the temperature calculation because of model geometry variation, different dimensions will be considered to study the influence of each one: distance between the wood centers (D), height of the wood column (H), width of the wood column (W), and thickness of the gypsum board (Tg). These geometric parameters were chosen because some frame wall builders and researchers propose recommended values [11,17,18]. Related to thermal effects, the distance between the wood centers can be interesting to study, to verify the thermal bridge effect and allow space for insulation. The greater width of the wooden column allows for a delay in the heating effect, as does the height dimension. The thickness of the gypsum board is an important factor in verifying the level of protection, as after the cladding falls off, charring increases at a much higher rate than that of initially unprotected wood.

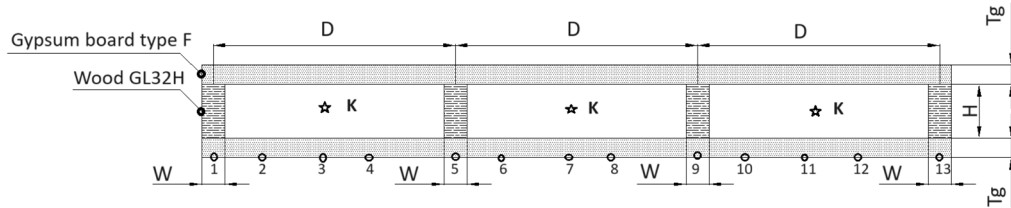

**Figure 4.** Two-dimensional hollow core element, with measuring points to the thermal insulation criterion.

Therefore, the present study focuses on determining the temperature range at the control points to obtain the fire resistance of the models under analysis. Concerning the studies, an iterative and sequential methodology was adopted, with each study consisting of four geometric parameter variations (D, H, W, and Tg), as represented in Table 3, which consists of determining the value relating to the geometric parameter that allows for a better performance to be obtained, that is, greater fire resistance.

**Table 3.** Numerical models.

| Geometric Parameters in the Study | Dimension [H × W + Tg] × D | | |
|---|---|---|---|
| D1 | [70 × 30 + 12.5] × 400 | [70 × 30 + 12.5] × 500 | [70 × 30 + 12.5] × 600 |
| H1 | [70 × 30 + 12.5] × D1 | [90 × 30 + 12.5] × D1 | [120 × 30 + 12.5] × D1 |
| W1 | [H1 × 30 + 12.5] × D1 | [H1 × 40 + 12.5] × D1 | [H1 × 50 + 12.5] × D1 |
| Tg1 | [H1 × W1 + 12.5] × D1 | [H1 × W1 + 15] × D1 | [H1 × W1 + 25] × D1 |

The study consists of the development of nine computational analyses using the finite element ANSYS® program [7]. When the effect of one of the geometric parameters is obtained in three of the models under analysis (in each line of Table 3), one of these results will immediately be used to control the next variable under study. In other words, in the variables analyzed in the following lines of Table 3, only two numerical simulations will always be analyzed. For each numerical model, the temperature field at control points and the respective fire resistance were obtained through the application of the thermal insulation criterion by EN1363-1:2020 [5], as represented in Figure 5.

According to the results, when D increases by 200 mm, the fire resistance decreases by only 1.6 min. This value is not representative (an average of 0.008 min/mm), and the option was to keep the dimension smaller to provide a smaller computational model. The fire resistance influenced by the dimension is not linear, but considering the limit values for each geometric parameter, it is possible to obtain an average value. For the other dimensions, when H increases by 50 mm, the fire resistance is 5.5 min (0.11 min/mm); if W increases by 20 mm, the fire resistance is 3.2 min (0.16 min/mm); and if Tg increases by 12.5 mm, the fire resistance is 174.6 min (13.97 min/mm).

Also, it will be possible to understand the influence of variables in the calculation of fire resistance. The best performance was obtained when D was equal to 400 mm, H was equal to 120 mm, W was the equivalent of 50 mm, and Tg was 25 mm. According to the results, it will be possible to understand the influence of the variables in the calculation of fire resistance, as shown in Figure 6.

The results show the temperature profile in all models with the best previously obtained variable to increase fire resistance. In all models, the exposed side reaches values above 950 °C when calculating fire resistance.

The effect of the distance between the wood centers (D = 400) and the height of the wood columns (H = 120) allows us to obtain a very close resistance to fire, around 75 min. Furthermore, in models [70 × 30 + 12.5] × 400 and [120 × 30 + 12.5] × 400, the lowest temperature is close to 72 °C.

In the model [120 × 50 + 12.5] × 400, with the effect of the width of the wood column (W = 50), there is an increase in fire resistance to 78.9 min compared to the previous one. The minimum temperature in the wood columns decreases by half, 40 °C.

In the model [120 × 50 + 25] × 400, an increase in double the thickness of the gypsum board is the best solution, with 253.5 min of fire resistance. Fire resistance increases more than three times compared to other variable geometric effects. Part of the wood columns remain at a temperature of 62 °C.

*4.2. Temperature Evolution Inside the Hollow Core Element*

For previous models which showed better performance, and inside the hollow core element, the temperature evolution was also evaluated. Figure 7 represents the nodal temperature obtained in the middle of the cavity for each model, denoted as nodal point K. In all three cavities of each model, due to the boundary conditions, temperature has the same evolution.

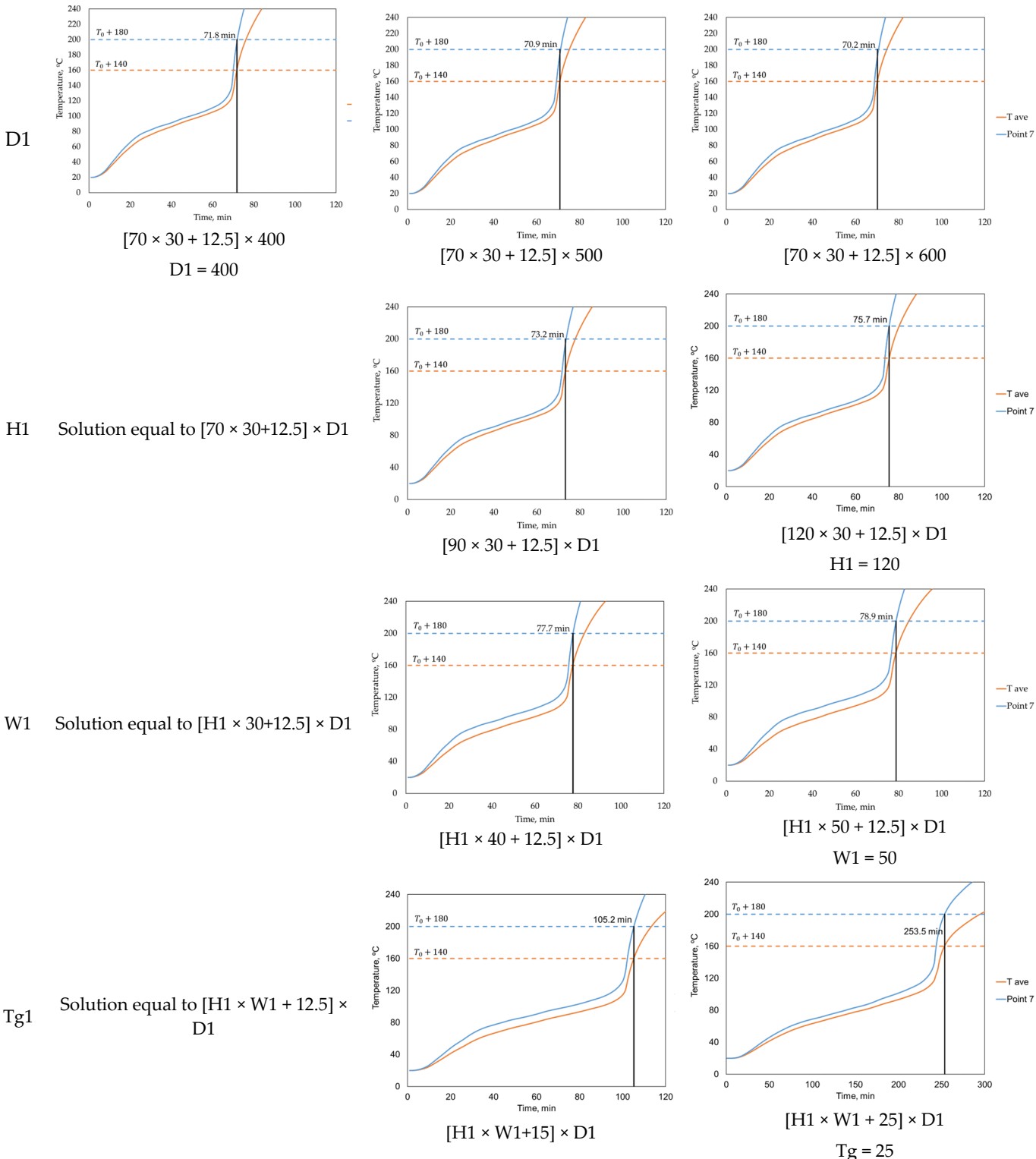

**Figure 5.** Temperature field at the application of the thermal insulation criterion in each model.

Model with higher performance           Temperature field, at the time of fire resistance

[70 × 30 + 12.5] × 400
Time of fire resistance = 71.8 min

[120 × 30 + 12.5] × 400
Time of fire resistance = 75.7 min

[120 × 50 + 12.5] × 400
Time of fire resistance = 78.9 min

[120 × 50 + 25] × 400
Time of fire resistance = 253.5 min

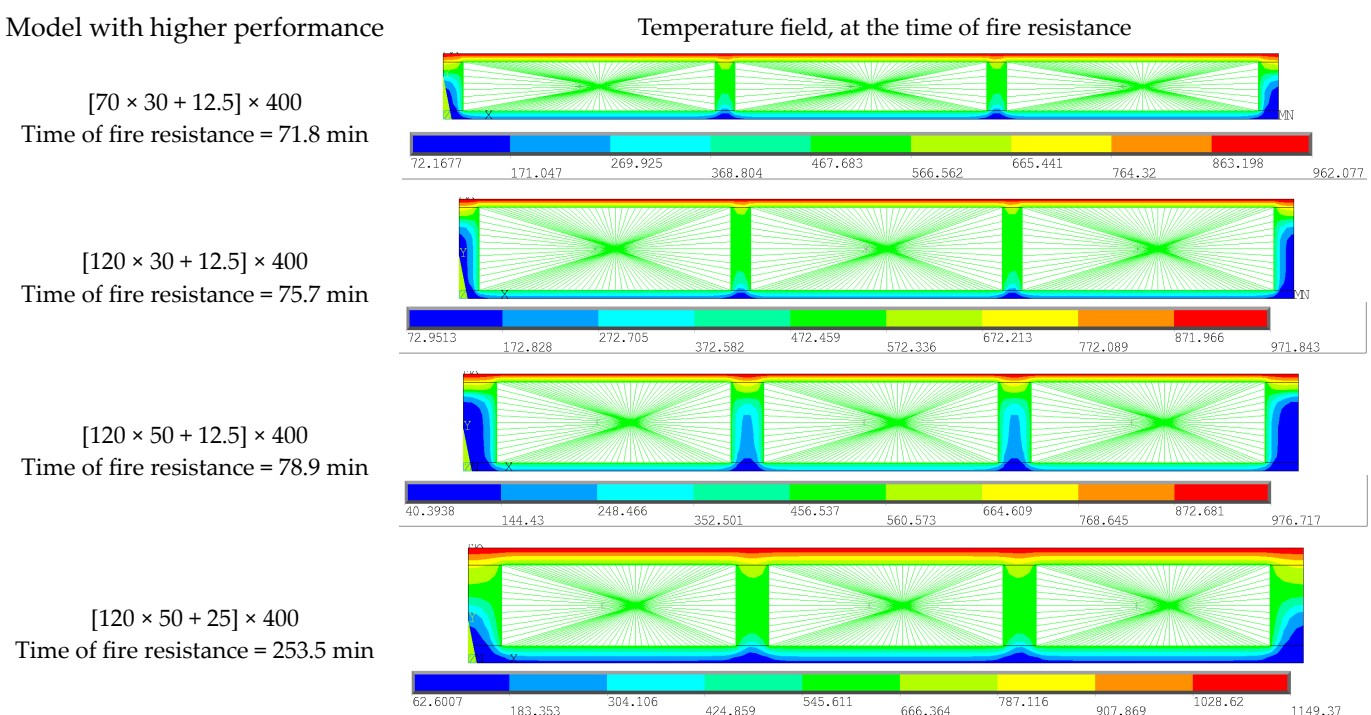

**Figure 6.** Temperature field at the time of fire resistance.

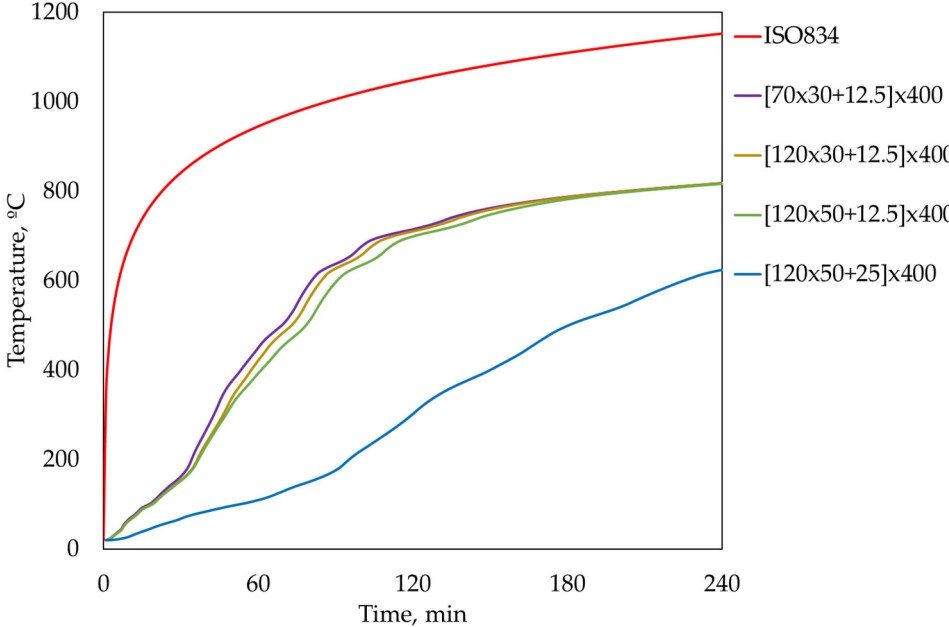

**Figure 7.** Temperature field in the inner hollow element.

These results show the temperature evolution during 240 min, after the time of fire resistance, for each model chosen with the best sized parameter. The main objective is to allow for the verification of the increase in heat inside the hollow core element during the time of exposure to fire. It is possible to compare the temperature obtained inside the cavity with the standard fire curve ISO834 [16]. The validated numerical model allows for the calculation of the temperature of the inner hollow elements. The effect of double the gypsum boar thickness is very important, representing a great increase in resistance to fire. The parameter variation related to the distance between wood centers D, height H, and width W does not have a major influence. Nevertheless, as mentioned previously, the effect of W is more relevant than the others.

## 5. Conclusions

In this work, a numerical method was used to determine the temperature evolution in hollow core wood elements under fire. The analysis uses a thermal and transient nonlinear material algorithm with three different finite elements. With this methodology, it was possible to obtain fire resistance by applying the thermal insulation criterion. The results also allow us to investigate the effect of geometric parameters on hollow core wooden elements that contribute to increasing fire resistance.

The practical importance of the present study is related to the use of lightweight wooden constructive elements with a hollow core. This type of element is normally constructed with prefabricated wall panels to be used in different types of buildings. The prefabricated element modules are supplied with one or two board layers of insulation that provide different periods of fire resistance. In this work, for the typical material chosen, increasing the thickness of the gypsum board, an average of 13.97 min/mm of fire protection was obtained. The boards contribute to the fire protection of the constructive solution, although different types of protective materials can provide different levels of protection. Another important use of this constructive solution is the hollow core section, where protective materials can be used internally, both as acoustic and thermal insulation, with noncombustible characteristics. Accordingly, light wooden constructive elements are designed to be resistant to fire situations.

Future work will include the analysis of new constructive models with the effect of steel connectors between the assembled elements and the analysis of hollow core wood filled with insulation. As mentioned in [2], hollow core elements have better mechanical and thermal properties and have the potential to increase the energy efficiency of building envelopes, namely increasing fire resistance in these types of construction models. With our models, it was possible to confirm that this type of element allows for better performance if it has the chosen geometric parameters.

**Author Contributions:** Conceptualization, E.M.M.F.; methodology, M.O.; validation, D.P.; investigation, D.P.; writing—original draft preparation, E.M.M.F.; writing—review and editing, E.M.M.F. and M.O.; supervision, E.M.M.F. All authors have read and agreed to the published version of the manuscript.

**Funding:** This research received no external funding.

**Conflicts of Interest:** The authors declare no conflicts of interest.

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
