# Peer review of "Temperature Evolution inside Hollow Core Wood Elements and Fire Resistance"

_fire, doi:10.3390/fire7020057_

Round 1

Reviewer 1 Report

Comments and Suggestions for Authors

The article presents a numerical study on thermal behaviour of panel built with glulam and gypsum board.

In consideration for the applicability of the type of the wood structure aproached and the relevance of proposed analysis, the article can be improved in some aspects:

- It is important to write the thermal properties of the materials used in the numerical models instead of only indicating the references.

- Lines 93 to 100: it was mentioned that the procedure to create mesh was carried out in a way to keep the finite elements dimension less than 10 mm, based on a gypsum board thickness of 32 mm. But this thickness is different of the models presented.

- Comparing figure 6 and figure 7 (about the fourth model) the referred temperature field is apparently different. It is worthy to review such result.

- The Authors mentioned that the numerical modelling was validated by means of reference [8], which was recently submitted, but not published yet. It is important to state such validation procedure, showing a comparison of results. It is not possible, based on the content of the article, to certificate if the presented results are reliable.

Comments on the Quality of English Language

Regarding to writing quality, it is recommended to review the verb tense and the pronoun setting.

Author Response

Reviewer 1

  1. Comments and Suggestions for Authors

The article presents a numerical study on thermal behaviour of panel built with glulam and gypsum board.

In consideration for the applicability of the type of the wood structure aproached and the relevance of proposed analysis, the article can be improved in some aspects:

ANSWER – The authors thank the reviewer for his time and detailed review and constructive comments and suggestions. Changes to the article have been identified and highlighted in blue.

- It is important to write the thermal properties of the materials used in the numerical models instead of only indicating the references.

ANSWER – The authors thank the reviewer. The thermal material properties were included in the work maintaining all references needed, with detailed information in Tables 1 and 2.

- Lines 93 to 100: it was mentioned that the procedure to create mesh was carried out in a way to keep the finite elements dimension less than 10 mm, based on a gypsum board thickness of 32 mm. But this thickness is different of the models presented.

ANSWER – This sentence is about our previously developed and tested numerical model, which was adopted in this work. Nevertheless, we agree with this observation and in agreement with the new geometric models we present each mesh adopted with the same rule in Figure 2. The mesh size was determined considering the thin side, that is, the gypsum board with a thickness equal to 12.5 mm or 15 mm. In these zones, the number of finite elements was adjusted to two (less than 10 mm), always considering a mesh automatically generated by ANSYS® [7] with a size of 10 mm. For the gypsum board, with a thickness of 25 mm, 3 finite elements were generated.

- Comparing figure 6 and figure 7 (about the fourth model) the referred temperature field is apparently different. It is worthy to review such result.

ANSWER – The authors appreciate this analysis very much. Figure 6 was changed because the presented postprocessor results were for a different time instant and not for the time of fire resistance, which will be the main reason for the comparison. Now, new images in Figure 6 were obtained and corrected in the manuscript to agree with Figure 7. Thank you.

- The Authors mentioned that the numerical modelling was validated by means of reference [8], which was recently submitted, but not published yet. It is important to state such validation procedure, showing a comparison of results. It is not possible, based on the content of the article, to certificate if the presented results are reliable.

ANSWER – The authors thank the reviewer. The recently submitted reference was suppressed and others included. Nevertheless, additional information with two works from the authors was presented in the Introduction. To compare and validate our numerical model, previously in other publications, the authors have described all different numerical models that have been developed and compared with experimental tests (Test 4 and Test 6) presented by Takeda and Mehaffey.

Reviewer 2 Report

Comments and Suggestions for Authors

Presented paper can be published. However I suggest following improvements.

Data obtained by mathematical model (obtained by ANSYS) should be compared with data obtained by experiment. Because manuscript bring only data obtained by math model without comparison with experimental data. Second alternative is validation of obtained results by other way (as experimental data).  

Author Response

Reviewer 2

Comments and Suggestions for Authors

Presented paper can be published. However I suggest following improvements.

Data obtained by mathematical model (obtained by ANSYS) should be compared with data obtained by experiment. Because manuscript bring only data obtained by math model without comparison with experimental data. Second alternative is validation of obtained results by other way (as experimental data).

ANSWER – The authors thank the reviewer for the time spent and the review, as well as for the constructive comments made. To compare and validate our numerical model, previously in other published works, the authors have described all different numerical models that have been developed and compared with experimental tests (Test 4 and Test 6) presented by Takeda and Mehaffey. The authors include this information with the references in the Introduction. Changes to the article have been identified and highlighted in blue.

Reviewer 3 Report

Comments and Suggestions for Authors

This work uses a numerical method that combines a thermal and transient non-linear material algorithm with three different finite elements to determine the temperature evolution in hollow core wood element under fire. Overall, this paper provides some new insights on façade fire safety and is well written. The paper may be accepted by addressing the following comments/suggestions:

1. Table 1: Please clarify the rationality of the selected geometric parameters in the model.

2. In this paper, the effect of the variables (D, W, H, Tg) on the fire resistance of the material is considered respectively, and the optimal value of one of the variables is obtained by controlling the other three variables. But the change of four variables will change the material structure, is it possible to consider the coupling effect between them? For example, the simulation shows that the D value performs best at the selected minimum value of 400, while the simulation of W shows that it is optimal at the maximum of 50, which will directly compress the cavity spacing, does this mean that the smaller the cavity gap is safer? This seems inconsistent with the conclusion that "hollow core elements have better mechanical and thermal properties and have the potential to increase the energy efficiency of building envelopes) ".

3. The analysis of the simulation results in this paper is limited to the impact of each parameter on the results, and it is suggested to add some more in-depth analysis (such as what is the practical significance of this study for the safety improvement of such building form?).

4. It is suggested to unify the ranges of vertical coordinates in Figure 5.

Comments on the Quality of English Language

Moderate editing of English language required

Author Response

Reviewer 3

This work uses a numerical method that combines a thermal and transient non-linear material algorithm with three different finite elements to determine the temperature evolution in hollow core wood element under fire. Overall, this paper provides some new insights on façade fire safety and is well written. The paper may be accepted by addressing the following comments/suggestions:

ANSWER – The authors thank the reviewer for his time and detailed review, constructive comments, and suggestions to improve our manuscript. Additional introductions have been introduced in the manuscript highlighted in blue, to try to answer all your requests.

  1. Table 1: Please clarify the rationality of the selected geometric parameters in the model.

ANSWER –These geometric parameters were chosen because some frame wall builders and researchers propose recommended values. In the rewritten manuscript, the authors included this information with three references. Related to thermal effects, the distance between the wood centers can be interesting to study, to verify the thermal bridge effect and allow space for insulation. The greater width of the wooden column allows for a delay in the heating effect, as does the height dimension. The thickness of the gypsum board is an important factor in verifying the level of protection, as after the cladding falls off, charring increases at a much higher rate than that of initially unprotected wood.

  1. In this paper, the effect of the variables (D, W, H, Tg) on the fire resistance of the material is considered respectively, and the optimal value of one of the variables is obtained by controlling the other three variables. But the change of four variables will change the material structure, is it possible to consider the coupling effect between them? For example, the simulation shows that the D value performs best at the selected minimum value of 400, while the simulation of W shows that it is optimal at the maximum of 50, which will directly compress the cavity spacing, does this mean that the smaller the cavity gap is safer? This seems inconsistent with the conclusion that "hollow core elements have better mechanical and thermal properties and have the potential to increase the energy efficiency of building envelopes) ".

ANSWER – The authors thank the reviewer and agree with this analysis. The authors verified that when D increases by 200 mm, the time resistance decreases by only 1.6 minutes. The authors considered this value not representative (in mean 0.008 min/mm), and the option was to keep the dimension smaller, to provide a smaller computational model. The time resistance influenced by the dimension is not linear, but considering the limit values for each geometric parameter it is possible to obtain a mean value. For the other dimensions: when H increases by 50 mm the time resistance is 5.5 minutes (0.11 min/mm); if W increases by 20 mm the time resistance is 3.2 minutes (0.16 min/mm); and if the Tg increases by 12.5 mm the fire resistance time is 174.6 minutes (13.97 min/mm). Accordingly, the authors rewrote the manuscript to clarify this aspect.

  1. The analysis of the simulation results in this paper is limited to the impact of each parameter on the results, and it is suggested to add some more in-depth analysis (such as what is the practical significance of this study for the safety improvement of such building form?).

ANSWER – The authors expanded the conclusions of the manuscript and justified the practical importance of the present study related to the use of lightweight wooden constructive elements with a hollow core. This type of element is normally constructed with prefabricated wall panels to be used in different types of buildings. The prefabricated element modules are supplied with one or two board layers of insulation that provide different periods of fire resistance. In this work, for the typical material chosen, increasing the thickness of the gypsum board, an average of 13.97 min/mm of fire protection was obtained. The boards contribute to the fire protection of the constructive solution, although different types of protective materials can provide different levels of protection. Another important use of this constructive solution is the hollow core section, where protective materials can be used internally, both as acoustic and thermal insulation, with non-combustible characteristics. Accordingly, light wooden constructive elements are designed to be resistant to fire situations.

  1. It is suggested to unify the ranges of vertical coordinates in Figure 5.

ANSWER – The authors thank the reviewer. The vertical coordinates of the graphic were fixed.

Round 2

Reviewer 1 Report

Comments and Suggestions for Authors

The reviewer appreciated the revised version as welll as the answers provided by the Authors, it was possible to have a better comprehension of the main objective ( lines 45 to 60).

On the validation analysis: It is worth mentioning that it is recommended to show the validation analysis of the numerical model in the article, together with the main results. However, it was understood that this is not the main idea of ​​the presented work, then the authors chose to mention it in the bibliographic reference, which was also suitable and possible to be consulted.